



# Should seasonal rainfall forecasts be used for flood preparedness?

Erin Coughlan de Perez[1,3,4], Elisabeth Stephens[2], Konstantinos Bischiniotis[3], Maarten van Aalst[1,4], Bart van den Hurk[5], Simon Mason[4], Hannah Nissan[4], Florian Pappenberger[6]

[1]Red Cross Red Crescent Climate Centre, The Hague, 2521 CV, The Netherlands
[2]School of Archaeology, Geography and Environmental Science, University of Reading, Reading, RG6 6AH, United Kingdom
[3]Institute for Environmental Studies, VU University Amsterdam, 1081 HV, The Netherlands
[4]International Research Institute for Climate and Society, Columbia University, New York, 10964, USA
[5]Royal Netherlands Meteorological Institute (KNMI), De Bilt, 3731 GA, Netherlands
[6]European Centre for Medium-Range Weather Forecasts, Reading, RG2 9AX, United Kingdom

*Correspondence to*: Erin Coughlan de Perez (coughlan.erin@gmail.com)

**Abstract.** In light of strong encouragement for disaster managers to use climate services for flood preparation, we question whether seasonal rainfall forecasts should indeed be used as indicators of the likelihood of flooding. Here, we investigate the

primary drivers of flooding at the seasonal timescale across sub-Saharan Africa. Given the sparsity of hydrological observations, we input bias-corrected reanalysis rainfall into the Global Flood Awareness System to identify seasonal indicators of floodiness. Results demonstrate that in wet climates, even a perfect tercile forecast of seasonal total rainfall would provide little to no indication of the seasonal likelihood of flooding. The number of extreme events within a season shows the highest correlations with floodiness consistently across regions. Otherwise, results vary across climate regimes:

floodiness in arid regions in Southern and Eastern Africa shows the strongest correlations with seasonal average soil moisture and seasonal total rainfall. Floodiness in wetter climates of West and Central Africa and Madagascar shows the strongest relationship with measures of the intensity of seasonal rainfall. Measures of rainfall patterns, such as the length of dry spells, are least related to seasonal floodiness across the continent. Ultimately, identifying the drivers of seasonal flooding can be used to improve forecast information for flood preparedness, and avoid misleading decision-makers.

## 25   1 Introduction

Humanitarians have been investing significant attention and resources in the uptake and use of climate services to inform their work in disaster risk management. For example, disaster managers regularly participate in Regional Climate Outlook forums and climate service partnerships (Hewitt et al., 2012; ICPAC, 2016; Mwangi et al., 2014). While many early warning systems focus on short-term hydrological flood warnings, these climate service initiatives promote the use of forecasts of

seasonal total rainfall. The use of such forecasts have yielded mixed results when used to prepare for heightened flood risk in Africa, such as prepositioning flood relief items (Braman et al., 2013) and evacuating vulnerable people (Anon, 2016). In this article we question whether seasonal rainfall forecasts have been over promoted for their usefulness in flood preparation.





To clarify whether seasonal total rainfall forecasts indeed indicate increased risk of flooding, we identify the dominant drivers of seasonal flooding in different locations of sub-Saharan Africa. In many locations, it is likely that total rainfall is not the dominant driver, and other seasonal descriptors would give a better indication of the risk of flood hazards. Cumulative rainfall is not the dominant flood-generating process for floods in most river basins in the United States (Berghuijs et al., 2016a), and monthly total rainfall has not been shown to be a good indicator of regional river "floodiness", or the percentage of regional rivers with extreme flooding (Stephens et al., 2015).

In the context of sub-Saharan Africa, we quantify the relationship between seasonal total rainfall and floodiness, and explore whether there might be alternative variables with a stronger relationship to floodiness at the seasonal level. In each river basin, the catchment size and the climate regime will affect the influence of hydraulic routing, soil dynamics, and precipitation patterns; we therefore identify which hydrometeorological variables are most related to seasonal flood risk in each location. We investigate seasonal total rainfall as well as 14 other variables and their combinations for their relation to seasonal percentage floodiness.

## 2 Methods

Given the scarcity of hydrological data available for Africa, we offer an alternative methodology to that used by Berghuijs et al. (Berghuijs et al., 2016a) for assessing the dominant flood-generating mechanisms in a region. Rainfall estimates from ERA-interim Land (Balsamo et al., 2015) are used to force the Global Flood Awareness System, a global hydrological model (Alfieri et al., 2013). We calculate anomaly correlations between rainfall input and the predicted flooding, which is defined as the proportion of river cells that have extreme discharge in a region in a given time period (Stephens et al., 2015). We repeat this analysis with the 14 alternative variables, and develop a generalized linear model to identify which combinations of variables provided the greatest indication of flood hazard in each region.

Our methodology depends on the reanalysis for a climatology of rainfall and focuses on the hydrological model to estimate the consequences of this rainfall on river flows. This approach is not limited by a patchy observational network, and results can be compared across regions to inform regional policies. While the rainfall has been bias-corrected with observations, we would encourage the replication of this methodology using local rainfall observations for more detailed study of the local drivers of floodiness.





## 2.1 Rainfall

To calculate the rainfall indices, we use daily gridded reanalysis rainfall estimates from 1980 – 2010. The rainfall estimates are 24-hour totals from the ERA-Interim Land reanalysis, which is adjusted from ERA Interim calibrated using GPCP v2.1 data (Balsamo et al., 2015). Due to patchy observational networks, uncertainties in precipitation datasets over Africa are large (Sylla et al., 2013), and this bias correction was shown to improve the performance of river discharge simulations from ERA-Interim Land over Africa (Balsamo et al., 2015). The soil moisture estimates are also taken from the ERA-Interim Land dataset.

The area of study we have selected is sub-Saharan Africa, 16N – 35S, 17W – 52E. Because flooding primarily happens during the wet seasons, we applied a dry mask by eliminating all 3-month seasons that have an average of less than 15% of the total annual rainfall and also less than 50cm of rainfall in that season (Mason et al., 1999). To calculate seasonal total rainfall, we sum the daily rainfall estimates for each overlapping 3-month season (JFM, FMA, etc.) over a 2.5 degree gridbox, as this is the resolution of many seasonal forecasting products from the Global Producing Centres for Long-Range Forecasts (Barnston et al., 2003; WMO, n.d.).

## 2.2 Flooding

We use daily rainfall from ERA-Interim Land to drive a hydrological model to estimate river discharge. The system used here is the Global Flood Awareness System (GloFAS), which is comprised of a HTESSEL land surface model to generate surface and subsurface runoff and a Lisflood model to complete the routing and groundwater flows at a 0.1 degree resolution for the entire global land surface (Alfieri et al., 2013). In this study we focus on river flooding only, therefore we only consider GloFAS river gridpoints which have greater than 1000km2 upstream basin area. These river pixels are aggregated to the 2.5 degree resolution to match the rainfall scale.

There are several ways to define whether a location experienced "flooding", which is the variable of interest to the disaster manager. Here, we define flooding according to the return period of the discharge, such that extreme floods happen at approximately the same frequency throughout the study area. We focus on the 1 in 5 and 1 in 50 year events; these return periods are defined by fitting a Gumbel extreme value distribution to the daily flows (Alfieri et al., 2013).

To understand the magnitude of flooding in a 2.5 degree gridbox, we calculate "floodiness" as defined in Stephens et al (2015). Percentage floodiness is the percent of river pixels that have at least one day of flooding above the return period, and duration floodiness is the number of pixel-days that have flooding during that season. Our results were very similar between percentage and duration floodiness, therefore duration floodiness is not shown here.



## 2.3 Predictor variables

While seasonal total rainfall has demonstrated some predictability in this part of the world (Barnston et al., 2010; Weisheimer and Palmer, 2014), there are other variables that might be predicted at the seasonal level: frequency of extreme
5  events within a season, sub-seasonal rainfall patterns, soil moisture, and rainfall intensity. Here, we investigate whether variables in each of those categories could serve as a better indicator of flood risk in sub-Saharan Africa. In addition to seasonal total rainfall, we calculated 14 predictor variables at the seasonal level. These are defined as follows:

*Extreme events within a season:*

| 1 Day Above 95th | Number of days in the season during which daily precipitation is greater than the 95th percentile of daily precipitation of the entire timeseries |
| --- | --- |
| 1 Day Above 99th | Number of days in the season during which daily precipitation is greater than the 99th percentile of daily precipitation of the entire timeseries |
| 3 Days Above 75th | Number of 3-day events in the season during which 3-day precipitation is greater than the 75th percentile of 3-day precipitation of the entire timeseries |
| 3 Days Above 99th | Number of 3-day events in the season during which 3-day precipitation is greater than the 99th percentile of 3-day precipitation of the entire timeseries |
| 5 Days Above 99th | Number of 5-day events in the season during which 5-day precipitation is greater than the 99th percentile of 5-day precipitation of the entire timeseries |

*Patterns of rainfall within a season:*

| Rainy days | Seasonal count of number of days in which daily precipitation is greater than 1mm (Sillmann et al., 2013) |
| --- | --- |
| Mean wet spell length | Average  length of all wet spells in that season, where a wet spell is defined as the length of consecutive days in which daily precipitation is greater than 1mm |
| Median dry spell length | Median length of all dry spells in that season, where a dry spell is defined as the length of consecutive days in which daily precipitation is less than 1mm |
| Dry spell autocorrelation | Spearman rank lag-1 autocorrelation of successive dry spell lengths (Schleiss and Smith, 2016) |
| 3 Day autocorrelation | Spearman rank lag-3 autocorrelation of daily rainfall amounts |

*Soil moisture and intensity:*



| Soil moisture | Volumetric soil water layer 1: Top soil layer 0-7cm. Average daily soil moisture for the season in kg/m3, |
|---|---|
| Intensity | Total seasonal rainfall divided by the number of rainy days (see definition above) |
| Contribution of extremes | Total rainfall falling in days of 95[th] percentile or higher, divided by total seasonal rainfall (Alexander et al., 2013) |
| Burstiness 15day | Burstiness as defined in (Schleiss and Smith, 2016): $\frac{\sigma_\mu - \mu}{\sigma_\mu + \mu}$ where $\mu$ is the average time between a specific amount of rainfall (interamount time), held at 15 days, and $\sigma$ is the standard deviation of interamount times |

**2.4 Comparison**

We examine whether anomalously high values of these variables correlate with greater floodiness. Using seasonal anomalies for each variable, we calculate the Spearman rank correlation between the rainfall anomalies and floodiness at every
5 gridpoint, as the data is not normally distributed. To assess our confidence in these results, we bootstrap the timeseries to generate 1000 replicates using a block bootstrap of 5 seasons. If less than 5% of the rank correlations of these bootstrapped replicates have an opposite sign as the original result, we have confidence in our result. Only results with this level of confidence are plotted in the figures.

Basin hydrology can also lead to complex relationships between rainfall and flooding. We therefore explore the correlation between basin-level rainfall with basin-level floodiness. We average the rainfall variable and floodiness variable across Food Producing Units (FPU) (Cai and Rosegrant, 2002), which are defined by a combination of hydrological basins and geopolitical regions and are therefore relevant for decision-making purposes. We apply a drymask for an entire FPU if more than half of the gridpoints in the FPU are in a dry season. With these aggregated results, we then apply the same correlation
methods as for the gridpoints above.

Lastly, we fit a generalized linear model to three of the predictor variables from different categories that showed improvements in correlation relative to seasonal total rainfall. For the dependent variable, we use a binary dataset indicating the occurrence or not of floodiness above the 50-year return period. The model uses a binomial distribution with a logit link,
and uses 10-fold cross-validation to fit the glm. We select the most parsimonious model within 1 standard error of the model with the minimum standard error, using the glmnet package for R (Friedman et al., 2010).





## 3 Results and Discussion

Three-month seasonal total rainfall anomalies show significant correlation with floodiness in several regions (Figure 1). The relationship is weakest in West and Central Africa, and also weakens as flood severity increases.

When the rainfall and floodiness are aggregated by FPU and then correlated, the correlations improve in almost all locations, suggesting that seasonal total rainfall forecasts for FPUs (Figure 1 c and d) might be of greater use than gridbox forecasts (Figure 1 a and b) as a predictor of flood hazard. Different regional forecast aggregations could also be explored to determine whether this can be further optimized.

While the correlations are significant in many regions, there is considerable variation in floodiness that remains unexplained by this variable. To demonstrate this, we calculate the probability of flooding (floodiness greater than 0) conditional on seasonal rainfall being in the top tercile of the distribution, which is the focus of many seasonal forecasts. Ultimately, even if a top-tercile rainfall forecast were given with 100% certainty, it would represent only a small increase in the probability of flooding relative to climatology (Figure 1e).

In Figures 2-4 we display results from three different sets of possible predictor variables. In Figure 2 we plot the anomaly rank correlations with floodiness for five different measures of extreme precipitation events within a season. None of these rainfall variables are a better predictor of floodiness in all locations (Figure 2 second row); however, the number of rain events above the 99th percentile (1-day, 3-day, and 5-day events) tend to outperform seasonal total rainfall in the areas of
20 West and Central Africa (where seasonal total rainfall had the weakest correlations; see Figure 1).

Next, we analyzed five different measures of rainfall patterns within a season, including the length of dry spells and wet spells. Apart from in isolated locations, these measures do not have coherently stronger correlations with floodiness than seasonal total rainfall (Figure 3).

The last set of variables we explored included soil moisture and several measures of seasonal rainfall intensity. Figure 4a shows that in most regions floodiness is not more strongly correlated with soil moisture than it is with seasonal total rainfall. In comparison, seasonal rainfall intensity shows a slightly higher correlation with floodiness across the continent (4b), defined as the total precipitation divided by the number of rainy days. Similarly, the percent of seasonal rainfall occurring in
the top 95th percentile days, here called the "contribution of extremes", shows higher correlations in the West and Central Africa region (4c). Both of these variables show less variation across Koppen climate regions, compared to seasonal total rainfall (Figure 1). Burstiness (Schleiss and Smith, 2016) of a 15-day interamount time (4d) does not show better correlations with floodiness than does seasonal total rainfall.

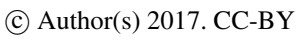



It is possible that a combination of these variables would outperform any of them in isolation, so we also test the combination of three different types of variables that each have strong correlations with floodiness: (1) 3 days above 99th, (2) soil moisture, and (3) contribution of extremes. To test whether a combination of these variables is better able to predict 50-year return period floodiness, we fit a logistic regression model for each gridpoint using these three variables. Because these variables are correlated with each other in several regions, we select the generalized linear model (glm) fit with the fewest variables that is still within one standard error of the optimal fitted model.

Results of the glm generally confirm the spatial patterns reflected in the correlation figures above, and indicate that a combination of these variables could be a useful indicator of floodiness in many regions. Figure 5 shows that the number of 3-day events above the 99[th] percentile was a meaningful contributor when added as a predictor independently, or in conjunction with another variable, in most of sub-Saharan Africa. Soil moisture is included as an additional predictor primarily in Southern Africa, while the contribution of extremes was included primarily in Central Africa. A combination of all three variables was recommended in East Africa and parts of Southern Africa, while none of the predictors was selected as a meaningful contributor for much of West and Central Africa.

# 4 Conclusions

In the analysis above, we have demonstrated that dominant flood-generating mechanisms differ widely across the African continent, using a methodology that can be replicated for other data-scarce regions to assess the key processes that cause flooding. Improvements both to the climatology of reanalysis rainfall and to the skill of global hydrological models could further improve the predictability of these processes, and we encourage replication of this methodology using observations to further describe and validate the flood-generating processes in specific locations.

It is clear that seasonal total rainfall is not a reasonable proxy for floodiness in most of West Africa, Central Africa, and Madagascar. These regions fall in the "equatorial" Koppen classification, which includes tropical savannahs. Floodiness in these regions demonstrated a stronger relationship with measures of the intensity of rainfall during a season than in the rest of the continent. In these regions, the climate services community should reconsider their association of seasonal total rainfall with flood risk and flood preparation measures (Braman et al., 2013). When using forecasts in an operational context, imperfect forecast skill of the rainfall proxy itself further reduces the usefulness of this information for flood preparedness.

On the other hand, much of East Africa, Southern Africa, and the Sahel are classified as "arid" in the Koppen climate classification, and these regions tend to show similar patterns in the dominant flood-generating mechanisms. Seasonal total rainfall had some of the highest correlations in these regions, as well as the number of extreme events within a season. These



findings are consistent with studies done in other arid areas. Berghuijs (2016) found that daily and multi-day rainfall events were the dominant flood-generating processes for river basins in arid regions of the United States, similar to the results in Figure 2d.

5 To maximize usefulness in these regions, forecasters could consider simple formatting alternatives to current forecasts that would provide a better indication of floodiness, such as replacing tercile forecasts with forecasts of the top percentiles of the distribution (Grieser, 2014), and offering aggregate forecasts for river basins or FPUs. The latter could also lend itself to greater forecast skill than for rainfall itself, and encourage regional-scale disaster preparedness.

10 Researchers developing new forecast products should consider several of the predictor variables discussed here. Forecasts of the frequency of extreme rainfall events would likely provide a better indication of floodiness, compared to seasonal total rainfall forecasts, for much of Sub-Saharan Africa. Studies have shown potential predictability of this variable in several locations (Anderson et al., 2015; Higgins et al., 2000; Verbist et al., 2010). Seasonal forecasts of soil moisture could give a useful indication of flood risk in dry regions of Africa (Figure 4), and these forecasts are also likely to have seasonal 15 predictability in areas where they can be well initialized, notably due to the persistence of soil moisture (Kanamitsu et al., 2002; Koster et al., 2010; Poveda et al., 2001). This also takes evaporation into account.

Forecasts of rainfall intensity could give a better indication of flood risk in West and Central Africa (Figure 5). However, intensity is the least spatially coherent and therefore least likely to be predictable (Moron et al., 2007). Further research into 20 the area is merited, as there are a few examples showing some potential predictability of rainfall intensity (Pineda and Willems, 2016).

Ultimately, the most informative forecasts of flood hazard at the seasonal scale are seasonal streamflow forecasts using hydrological models calibrated for individual river basins (Sahu et al., 2016). While this is more computationally and 25 resource intensive, investments in better forecasts of seasonal flood risk could be of immense use to the disaster preparedness community.

In their work, disaster managers can support these forecasting efforts by better defining the meteorological and hydrological variables that relate to disaster. Sharing this information with forecasters can inform the development of forecast products 30 that provide specific information about these "danger levels", thus better enabling stakeholders to take appropriate preparatory actions. Forecast-based finance initiatives are underway globally, with the aim to take action and release financing proportional to the risk information in a forecast, before the potential disaster (Coughlan de Perez et al., 2016). Changes to forecast products to provide clearer and more targeted risk information can support this process, and enable humanitarians to better anticipate and prepare for disasters before they strike.




**Data availability:** ERA Interim Land rainfall and soil moisture estimates that support the findings of this study are available from ECMWF http://apps.ecmwf.int/datasets/data/interim-land/type=an/ . GloFAS hydrological discharge estimates are generated from the Joint Research Centre and available in real time http://globalfloods.jrc.ec.europa.eu/. Derived data supporting the findings of this study are available from the corresponding author upon request.

**Acknowledgements**

We thank our colleagues for their insights and suggestions on indices to consider. We are grateful to the German Federal Foreign Office for their support to the development of Forecast-based Financing pilots around the world, which have inspired these research questions. This work was supported by the UK Natural Environment Research Council (NE/P000525/1). This work was also funded in part by grants/cooperative agreements from the National Oceanic and Atmospheric Administration (NA15OAR4310076 and NA13OAR4310184). The views expressed are those of the authors and do not necessarily reflect the views of NOAA or its subagencies. E. Stephens' time was funded by Leverhulme Early Career fellowship ECF-2013-492.



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

**Figure 1**: Anomaly rank correlations between seasonal total rainfall and percentage floodiness (Stephens et al., 2015) at the
5-year (a), and 50-year (b) return periods. Anomaly rank correlations between seasonal total rainfall for a 2.5 degree gridded
Food Producing Unit (FPU) and floodiness for that FPU at the 5-year (c) and 50-year (d) return periods. Correlations are
only shown here if more than 95% of all boostrapped replicates agreed on the sign of the result. The increase in probability
of floodiness above the 5-year return period conditional on seasonal total rainfall falling in the top tercile (e), expressed as
the difference in probability relative to climatology.





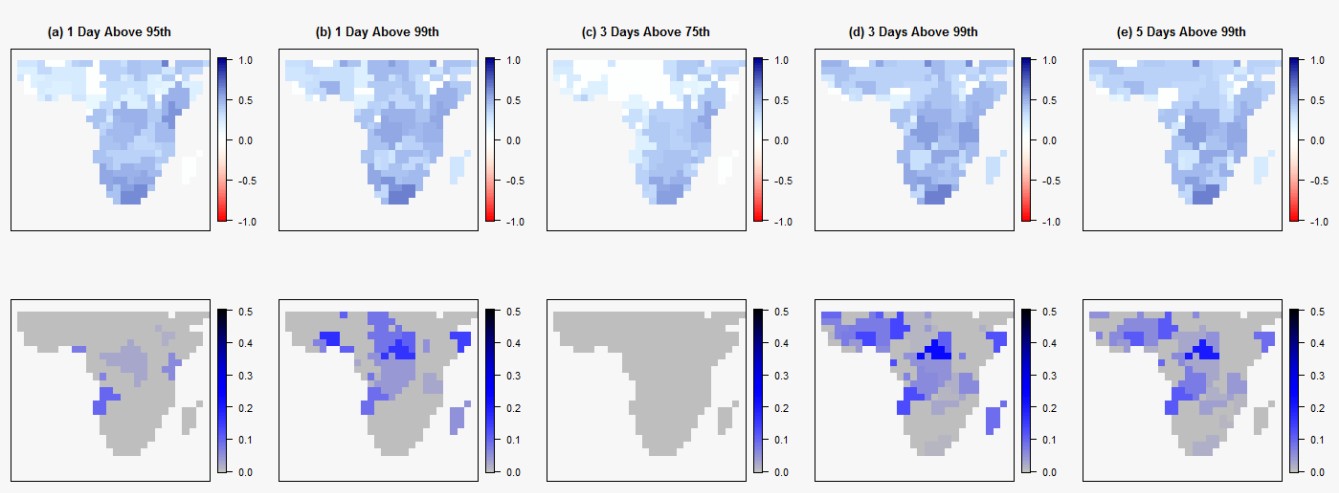

**Figure 2**: Correlation of number of extreme events within a season and floodiness for FPUs in Africa. The top row shows the anomaly rank correlations between each variable and percentage floodiness at the 5-year return period at the FPU level. The bottom row is the improvement relative to seasonal total rainfall – locations in blue show a higher anomaly correlation for this variable than for seasonal total rainfall anomalies. Areas in which seasonal total rainfall has a higher or equal correlation are shown in grey. Note that results are only plotted for locations where more than 95% of the boostrapped replicas agree on the sign of the change.

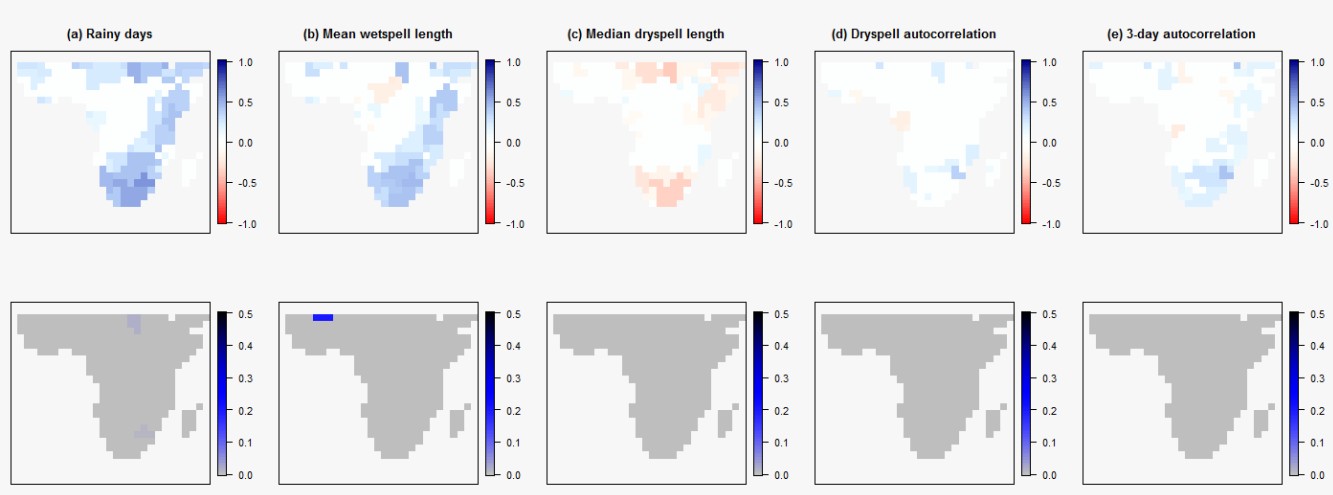

**Figure 3**: Same as figure 2 for the following variables (a) Rainy days: number of days with more than 1mm of rain (b) Mean wetspell length: mean length of consecutive days of rain greater than 1mm, (c) Median dryspell length: median length of consecutive dry days, (d) Dryspell autocorrelation: Spearman rank lag-1 autocorrelation of successive dry spell lengths, (e) 3-day autocorrelation: Spearman rank lag-3 autocorrelation of daily rainfall amounts.





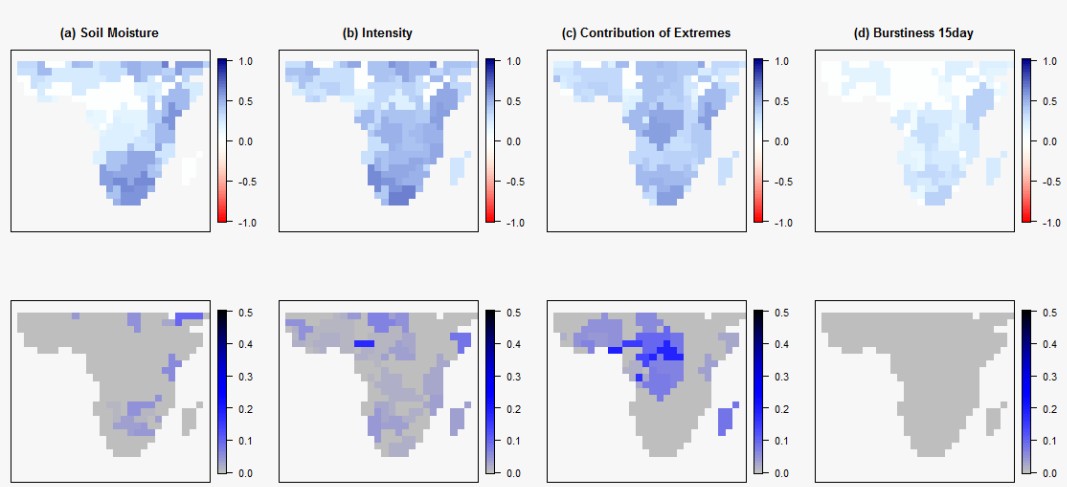

**Figure 4**. Same as figure 2 for the following variables (a) Soil Moisture: seasonal average moisture in topsoil (b) Intensity: total rainfall divided by the number of rainy days, (c) Contribution of Extremes: total rainfall divided by the amount of rain contributed by the top 95[th] percentile days, (d) Burstiness 15day: Intermittency measure (Schleiss and Smith, 2016).

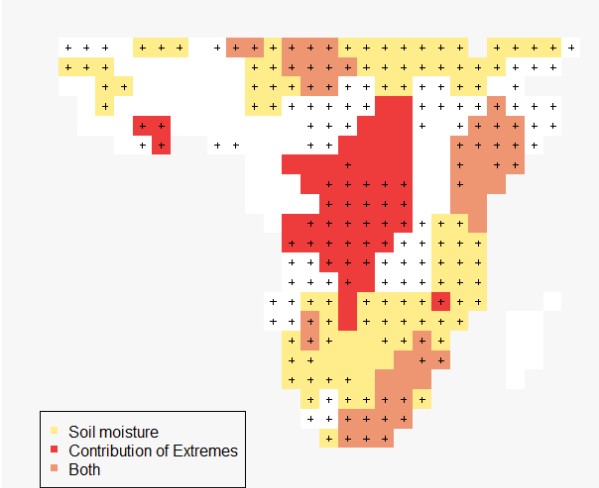

**Figure 5**. Results of optimizing a logistic regression model using a combination of the high-performing variables considered earlier. The model predicted whether there was any floodiness at the 50-year return period by using the following predictors: number of 3-day events in the 95[th] percentile (crosses), soil moisture (yellow), and the contribution of extremes (red). To

10    optimize the model, we selected the most parsimonious combination of these three predictors that formed a glm that is within one standard error of the standard error that could be achieved by the maximum fit. FPUs that are plain white showed no value in using any of the predictors, while locations with colors/symbols show which predictors were retained in the optimized model, either alone or in combination with other predictors.