# Peer review of "Should seasonal rainfall forecasts be used for flood preparedness?"

_Hydrology and Earth System Sciences, 2017_

## Referee Comment (RC1) · Anonymous Referee #1 · 13 Apr 2017

General comments:

The paper questions whether seasonal rainfall information can be used to indicate the likelihood of flooding within a season, focusing on sub-Saharan Africa. In particular the paper focuses on correlations between different seasonal rainfall variables (e.g. total seasonal rainfall, mean rainfall intensity and cumulative wet days) and "floodiness" determined through using a reanalysis dataset to drive a global hydrological model. The authors conclude that forecasts of seasonal total rainfall may be less informative than other more granular metrics, providing further motivation for studies to understand what seasonal forecast variables can best inform disaster management and humanitarian decisions.

This paper provides a concise and interesting research contribution on an important

topic with implications for disaster risk management and the design of seasonal climate services. It is well written, focused, and provides a balanced interpretation of the evidence provided through analysing reanalysis and hydrological model datasets. The paper will be of interest to those who are involved developing climate services, particularly using seasonal forecasts, humanitarian agencies, government decision makers addressing flood risks, and the climate scientists advancing methods for relating long-term rainfall patterns to the risk of flooding events. This paper will provide a valuable contribution to the literature.

Below are some relatively minor recommended changes that should help further improve the paper, focusing on refining some of the key arguments and explanation of the results.

Specific comments:

1) Abstract, line beginning "Results demonstrate...": the evidence of "little to no indication..." is not necessarily true of all wet climate regions in the study area and is perhaps an over-generalisation. Suggest rephrasing – i.e. some regions of west, central and east Africa with typically wet climates.

2) The term "flood-generating process" is used throughout the paper (e.g. in section 4) when referring to measures of seasonal rainfall and their correlations to "floodiness". I am not sure the terminology is entirely appropriate since the measures evaluated in this paper are statistical indicators/quantities as opposed to physical processes (e.g. convective or frontal rainfall). Consider revising this terminology to something less associated with processes – e.g. "Total seasonal rainfall is not a reliable indicator of the intensity of flood events within a season in most river basins...".

3) Last sentence of section 2.1: The horizontal resolutions of seasonal forecasting systems from global producing centres have increased substantially in recent years, and many operational systems now run at 0.5 degrees and sometimes as high as 0.25 degrees. The justification of using a 2.5 degree resolution therefore needs to be

revised, with reference to more recent literature (a paper from 2003 is currently cited).

4) Section 3, second sentence: In addition to West and Central Africa, from viewing the figures I would add the Greater Horn of Africa region in East Africa as a region where the relationship appears weak. The reference to West and Central Africa is mentioned in other places in the paper so check the consistency in the rest of the paper after making any revisions.

5) Section 4, second sentence: Insert "understanding of" before the word "predictability". The point being that predictability comes from the accuracy of the forecast models used to predict seasonal rainfall and not the quality of the reanalysis data per se.

6) Section 4, paragraph 2: The reference to Koppen climate classifications is first made here. Whilst I can see the value in linking the relationships between seasonal rainfall metrics and floodiness to different climate types, there is a risk of over-generalising the results. The climate types within East Africa and southern Africa (and elsewhere) vary greatly so to generalise by saying these regions are classified as "arid" is misleading – some areas are far from arid – and further using this as a basis to generalise the results of the study risks over-simplifying the findings. Understanding the robustness of these findings for different climate types would require further investigation.

Technical corrections:

1) Section 1, second paragraph, final sentence needs rephrasing to improve clarity.

2) Suggest inserting "many parts of" between "for" and "Africa" in first sentence of section 2.

3) Is the third predictor variable definitely at the 75th percentile? The results do seem consistent with this but just checking as the 1 day variable is 95th and 99th percentile whilst 3 day is 75th and 99th.

4) "Floodiness" is first defined in section 2.2 but used earlier in the paper. Either define this term earlier or state that it will be defined in section 2.2 when first introduced.

5) Section 2.2: I think it would be helpful to know approximately how many river pixels typically can be found within 2.5 degree gridbox. This would help in interpreting the "floodiness" metric when used throughout the paper.

6) Section 2.4, third paragraph – acronym GLM needs introducing earlier (not in section 3).

7) Section 3, fifth paragraph, consider rephrasing second sentence beginning "Figure 4a" to replace "not more strongly" – this is a little confusing.

8) The figures would benefit from latitude and longitude values on the axes.

---

## Referee Comment (RC2) · Anonymous Referee #1 · 13 Apr 2017

Please note that the points made under "specific comments" and "technical corrections" are separated by numbers. They were intended to be separated onto different lines but unfortunately have appeared in continuous lines in two paragraphs.

---

## Referee Comment (RC3) · Anonymous Referee #2 · 26 Apr 2017

The topic of the paper is important for practical applications. The research presented is of high quality, the paper is well written, and the methodology used is sound. I have only a few comments. (1) I have problems with some of the terms used in the paper, such as drivers of flooding, flood-generating processes and etc. The paper is not identifying the drivers or processes, but rather identifying proxies or indicators of floodiness through correlation analyses. (2) While there are a few comments in the paper on the skill (or lack of skill) of seasonal GCMs in forecasting the proxies (indicators), they are dispersed in discussion and conclusions. I would like to see a more focused discussion on this, including implications, given the already low correlation between proxies and floodiness. (3) I don't seem to be able to work out from the paper the source of the soil moisture data. (4) Given many seasonal GCMs also produce surface runoff, I wonder whether the authors would like to comment on the value of using surface runoff

forecasts from the GCMs. (5) Seasonal GCMs generally do a good job in forecasting large climate patterns (such as represented by SST based climate indices). It will be of value to add climate indices as predictors in analyses. It may well be that these will give the best correlations, especially when it is factored that GCMs are generally of low skill in forecasting climate variables directly.

---

## Editor Comment (EC1) · Q. ̆J. Wang (Editor) · 8 May 2017

Dear Authors

You will need to make initial replies to comments by the reviewers. After that, I will review both the reviewer comments and your replies and advise you on the next step. Thanks

Best regards

QJ Wang

---

## Author Comment (AC1) · 29 May 2017

COMMENT: General comments: The paper questions whether seasonal rainfall information can be used to indicate the likelihood of flooding within a season, focusing on sub-Saharan Africa. In particular the paper focuses on correlations between different seasonal rainfall variables (e.g. total seasonal rainfall, mean rainfall intensity and cumulative wet days) and "floodiness" determined through using a reanalysis dataset to drive a global hydrological model. The authors conclude that forecasts of seasonal total rainfall may be less informative than other more granular metrics, providing further motivation for studies to understand what seasonal forecast variables can best inform disaster management and humanitarian decisions.

This paper provides a concise and interesting research contribution on an important

topic with implications for disaster risk management and the design of seasonal climate services. It is well written, focused, and provides a balanced interpretation of the evidence provided through analysing reanalysis and hydrological model datasets. The paper will be of interest to those who are involved developing climate services, particularly using seasonal forecasts, humanitarian agencies, government decision makers addressing flood risks, and the climate scientists advancing methods for relating longterm rainfall patterns to the risk of flooding events. This paper will provide a valuable contribution to the literature.

Below are some relatively minor recommended changes that should help further improve the paper, focusing on refining some of the key arguments and explanation of the results.

RESPONSE: Thank you very much for this summary and your insightful comments. We indeed hope that this research contribution will be used by those who are developing climate services. We have addressed each of your comments below, and appreciate these improvements to the manuscript.

COMMENT: Specific comments: 1) Abstract, line beginning "Results demonstrate...": the evidence of "little to no indication..." is not necessarily true of all wet climate regions in the study area and is perhaps an over-generalisation. Suggest rephrasing – i.e. some regions of west, central and east Africa with typically wet climates.

RESPONSE: Thank you for the suggestion; we have implemented this change in the text.

COMMENT: 2) The term "flood-generating process" is used throughout the paper (e.g. in section 4) when referring to measures of seasonal rainfall and their correlations to "floodiness". I am not sure the terminology is entirely appropriate since the measures evaluated in this paper are statistical indicators/quantities as opposed to physical processes (e.g. convective or frontal rainfall). Consider revising this terminology to something less associated with processes – e.g. "Total seasonal rainfall is not a reliable

indicator of the intensity of flood events within a season in most river basins...".

RESPONSE: Indeed, you make a good distinction, which was also noted by reviewer 2. We have retained the phrase "flood-generating mechanisms" in the introduction when discussing the work of Berghuijs et al., as this is the terminology used by them for their work. In all other instances when referring to our own analysis, we have adjusted the terms used to clarify that we are examining statistical indicators. This includes in the methods section (line 18) and the conclusions section 4.

COMMENT: 3) Last sentence of section 2.1: The horizontal resolutions of seasonal forecasting systems from global producing centres have increased substantially in recent years, and many operational systems now run at 0.5 degrees and sometimes as high as 0.25 degrees. The justification of using a 2.5 degree resolution therefore needs to be revised, with reference to more recent literature (a paper from 2003 is currently cited).

RESPONSE: Indeed, while many seasonal forecasts are available at higher resolutions, there is a tradeoff between spatial structure and statistical significance. We do not believe that repeating the calculations at a higher resolution would provide more information, but rather introduce noise. In fact, the larger scale of FPUs showed a greater relationship between rainfall and floodiness. We do note that 2.5 degree resolution is the WMO standard for Global Producing Centres (GPCs) of Long-Range Forecasts.

COMMENT: 4) Section 3, second sentence: In addition to West and Central Africa, from viewing the figures I would add the Greater Horn of Africa region in East Africa as a region where the relationship appears weak. The reference to West and Central Africa is mentioned in other places in the paper so check the consistency in the rest of the paper after making any revisions.

RESPONSE: While it might appear that the Greater Horn has a weaker relationship, we would note that there are a few pixels that contain only water in the top right, which are grey. This might cause a perception that the region has a weaker relationship than

it actually does. We will clarify in the text.

COMMENT: 5) Section 4, second sentence: Insert "understanding of" before the word "predictability". The point being that predictability comes from the accuracy of the forecast models used to predict seasonal rainfall and not the quality of the reanalysis data per se.

RESPONSE: Excellent point; we have implemented this suggestion.

COMMENT: 6) Section 4, paragraph 2: The reference to Koppen climate classifications is first made here. Whilst I can see the value in linking the relationships between seasonal rainfall metrics and floodiness to different climate types, there is a risk of over-generalising the results. The climate types within East Africa and southern Africa (and elsewhere) vary greatly so to generalise by saying these regions are classified as "arid" is misleading – some areas are far from arid – and further using this as a basis to generalise the results of the study risks over-simplifying the findings. Understanding the robustness of these findings for different climate types would require further investigation.

RESPONSE: Noted; we agree that there is considerable variation in climate in each of these regions. We have adjusted the language accordingly.

COMMENT: Technical corrections:

RESPONSE: Thank you for these thoughtful corrections; we have noted below as they have been incorporated.

COMMENT: 1) Section 1, second paragraph, final sentence needs rephrasing to improve clarity.

RESPONSE: OK

COMMENT: 2) Suggest inserting "many parts of" between "for" and "Africa" in first sentence of section 2.

RESPONSE: OK

COMMENT: 3) Is the third predictor variable definitely at the 75th percentile? The results do seem consistent with this but just checking as the 1 day variable is 95th and 99th percentile whilst 3 day is 75th and 99th.

RESPONSE: Yes, that is correct.

COMMENT: 4) "Floodiness" is first defined in section 2.2 but used earlier in the paper. Either define this term earlier or state that it will be defined in section 2.2 when first introduced.

RESPONSE: OK

COMMENT: 5) Section 2.2: I think it would be helpful to know approximately how many river pixels typically can be found within 2.5 degree gridbox. This would help in interpreting the "floodiness" metric when used throughout the paper.

RESPONSE: Excellent idea. We will include a map of number of river pixels per gridbox.

COMMENT: 6) Section 2.4, third paragraph – acronym GLM needs introducing earlier (not in section 3).

RESPONSE: OK – we have added it at the beginning of that section.

COMMENT: 7) Section 3, fifth paragraph, consider rephrasing second sentence beginning "Figure 4a" to replace "not more strongly" – this is a little confusing.

RESPONSE: OK

COMMENT: 8) The figures would benefit from latitude and longitude values on the axes.

RESPONSE: OK

---

## Author Comment (AC2) · 29 May 2017

COMMENT: The topic of the paper is important for practical applications. The research presented is of high quality, the paper is well written, and the methodology used is sound. I have only a few comments.

RESPONSE: Thank you for your specific comments, and we have incorporated further discussion on some of the critical points you have mentioned below. We appreciate that you see the value of this work for practical applications.

COMMENT: (1) I have problems with some of the terms used in the paper, such as drivers of flooding, flood-generating processes and etc. The paper is not identifying the drivers or processes, but rather identifying proxies or indicators of floodiness through correlation analyses.

RESPONSE: Thank you for this point; indeed reviewer #1 also mentioned this, and we have adjusted the language accordingly.

COMMENT: (2) While there are a few comments in the paper on the skill (or lack of skill) of seasonal GCMs in forecasting the proxies (indicators), they are dispersed in discussion and conclusions. I would like to see a more focused discussion on this, including implications, given the already low correlation between proxies and floodiness.

RESPONSE: This lack of skill can point in several directions for further research, and we have added the following paragraph before the discussion of seasonal hydrological modeling:

Seasonal skill in forecasting total 3-month rainfall anomalies is varied around the world; highest skill has been achieved during ENSO events in areas that have ENSO teleconnections (Barnston et al., 2010a; Weisheimer and Palmer, 2014). Given the low correlations we have found here between floodiness and either seasonal total rainfall or other rainfall indicators, forecasts of any of these proxies are unlikely to provide strong signals of increased risk. However, there have been several studies using large-scale climate patterns and sea-surface temperatures (SSTs) as predictors of flood risk, most focusing on the role of ENSO in changing global flood risk (Emerton et al., 2017; Ward et al., 2014, 2016). Further research on using SSTs and other climate patterns to directly forecast changes to flooding is merited, to explore whether such forecasts would give stronger indications of change in flood hazard than seasonal climate models of rainfall.

COMMENT: (3) I don't seem to be able to work out from the paper the source of the soil moisture data.

RESPONSE: This is mentioned on page 3 line 7-8: they are taken from the ERA-Interim Land dataset.

COMMENT: (4) Given many seasonal GCMs also produce surface runoff, I wonder

whether the authors would like to comment on the value of using surface runoff forecasts from the GCMs.

RESPONSE: Indeed, this is a good point. We refrained from including surface runoff in this paper, as it is no longer a rainfall indicator, but rather a rudimentary hydrological model. In the conclusions section, you can see that we do suggest people consider developing and running different seasonal hydrological models to provide specific floodiness seasonal forecasts, but we have not attempted to identify and evaluate those models that exist.

However, we did indeed analyze runoff from the ERA-Interim Land dataset, and the results are promising, in that even a crude hydrological model can greatly improve the correlation with floodiness. In case of interest: you can see the results at the end of this comment as per Figure 2 in the manuscript.

COMMENT: (5) Seasonal GCMs generally do a good job in forecasting large climate patterns (such as represented by SST based climate indices). It will be of value to add climate indices as predictors in analyses. It may well be that these will give the best correlations, especially when it is factored that GCMs are generally of low skill in forecasting climate variables directly.

RESPONSE: This is an excellent suggestion, and we also agree that this is likely to hold promise. Ward et al. have developed maps of the relationship between ENSO indices and global river discharge as well as global flood frequencies and durations (P J Ward, Kummu, & Lall, 2016; Philip J. Ward, Beets, Bouwer, Aerts, & Renssen, 2010). Emerton et al. 2017 also produced a study on the complexity of the relationship between ENSO and flood hazard (Emerton et al., 2017), and there has also been work done to link climate patterns with water scarcity (Veldkamp, Eisner, Wada, Aerts, & Ward, 2015).

We agree that this merits further research, including analysis of climate patterns beyond ENSO itself, and analysis of the connection with different types of floodiness.

As it would be beyond the scope of this paper to do a comprehensive study of these connections, we will instead pursue your suggestion as a separate paper, which will complement the current discussion of seasonal forecasts and floodiness.

[Figure]

**Runoff**

Correlation color scale: 1.0, 0.5, 0.0, -0.5, -1.0

**Fig. 1.** Correlation of seasonal average runoff and floodiness for FPUs in Africa. These are anomaly rank correlations between runoff and percentage floodiness at the 5-year return period at the FPU level.

**Fig. 2.** The improvement relative to seasonal total rainfall – locations in blue show a higher anomaly correlation for this variable than for seasonal total rainfall anomalies.